# The Potential Association between E2F2, MDM2 and p16 Protein Concentration and Selected Sociodemographic and Clinicopathological Characteristics of Patients with Oral Squamous Cell Carcinoma

Agata Świętek [1,2], Karolina Gołąbek [1,*], Dorota Hudy [1], Jadwiga Gaździcka [1], Krzysztof Biernacki [1], Katarzyna Miśkiewicz-Orczyk [3], Natalia Zięba [3], Maciej Misiołek [3] and Joanna Katarzyna Strzelczyk [1]

[1] Department of Medical and Molecular Biology, Faculty of Medical Sciences in Zabrze, Medical University of Silesia in Katowice, 19 Jordana St., 41-808 Zabrze, Poland

[2] Silesia LabMed Research and Implementation Centre, Medical University of Silesia in Katowice, 19 Jordana St., 41-808 Zabrze, Poland

[3] Department of Otorhinolaryngology and Oncological Laryngology, Faculty of Medical Sciences in Zabrze, Medical University of Silesia in Katowice, 10 C Skłodowskiej St., 41-800 Zabrze, Poland

* Correspondence: kgolabek@sum.edu.pl

**Abstract:** Background: E2F transcription factor 2 (E2F2), murine double minute 2 (MDM2) and p16 are some of the key proteins associated with the control of the cell cycle. The aim of this study was to evaluate E2F2, MDM2 and p16 concentrations in the tumour and margin samples of oral squamous cell carcinoma and to assess their association with some selected sociodemographic and clinicopathological characteristics of the patients. Methods: The study group consisted of 73 patients. Protein concentrations were measured by enzyme-linked immunosorbent assay (ELISA). Results: There were no statistically significant differences in the levels of E2F2, MDM2 or p16 in the tumour samples as compared to the margin specimens. We found that patients with N0 showed significantly lower E2F2 concentrations than patients with N1 in the tumour samples and the median protein concentration of E2F2 was higher in HPV-negative patients in the tumour samples. Moreover, the level of p16 in the margin samples was lower in alcohol drinkers as compared to non-drinkers. Similar observations were found in concurrent drinkers and smokers compared to non-drinkers and non-smokers. Conclusions: E2F2 could potentially promote tumour progression and metastasis. Moreover, our results showed a differential level of the analysed proteins in response to alcohol consumption and the HPV status.

**Keywords:** E2F2; MDM2; p16; protein concentration; ELISA; oral squamous cell carcinoma; tumour; margin

## 1. Introduction

Oral squamous cell carcinoma (OSCC) accounts for approximately 90% of head and neck cancers (HNSCC) with nearly 400,000 new cases diagnosed worldwide in 2020 [1]. The known risk factors for the development of this type of cancer include primarily the mucosal exposure to tobacco smoke and alcohol. Other risk factors include air pollution, UV exposure and viral infections. Another potential group of risk factors is related to endogenous factors, such as genetic predisposition, including abnormalities in proliferation, apoptosis, differentiation, signal transduction, angiogenesis, cell cycle regulation and DNA repair [2–6]. E2F transcription factor 2 (E2F2), murine double minute (2 MDM2) and p16 (also known as cyclin-dependent kinase inhibitor 2A, CDKN2A) are some of the key proteins involved in the control of the cell cycle [7–9]. E2F2 is a member of the E2F family of proteins and plays a complex role in the body, mainly in the control of gene expression and hence, in the cell cycle control [10,11]. In addition, it is involved

in apoptosis, inflammation, migration and cell invasion. Some studies have shown that E2F2 can presumably promote tumour development and progression in various types of cancer, including lung adenocarcinoma, breast cancer, ovarian cancer and colorectal cancer [12,13]. MDM2 is an oncoprotein with the ability to negatively regulate and maintain stability of the p53 tumour suppressor protein signalling pathway [9]. MDM2 and p53 form an autoregulatory feedback loop in which p53 elevates the MDM2 levels, while MDM2 inhibits the expression and activity of p53 [14]. It is known that MDM2 overexpressed in many types of cancer (lung cancer, breast cancer, liver cancer, oesophagogastric cancer and colorectal cancer) can act as an oncogene [9,15,16]. Interestingly, the value of MDM2 as a prognostic marker remains unclear and dependent on the tumour type [17]. p16 is a tumour suppressor protein that plays an important function in the regulation of the cell cycle associated with the inhibition of the cyclin-dependent kinases (CDK4 and CDK6). It is essential in the inhibition of the cell transition from the G1 phase to the S phase of the cell cycle [18]. p16 cooperates with the retinoblastoma (Rb1) and the p53 tumour suppressor genes. Significant upregulation of p16 may result from inactivation of p53 or Rb1. This mechanism is strongly related to inactivation of Rb1 by the HPV E7 protein [19,20]. Not all mechanisms associated with p16 expression are known in cancers. Although many studies have reported on the prognostic role of the decreased or increased p16 expression, the results are often conflicting [20].

According to the "field cancerization" concept proposed by Slaughter et al. [21], histological changes are present in the epithelium surrounding a squamous cell tumour. This may indicate that, probably as a result of exposure to carcinogens, this area plays a key role in the development of numerous foci of malignant transformation, including second primary tumours after surgical intervention and locally recurrent tumours [21]. Many subsequent molecular discoveries have confirmed the proposed model of carcinogenesis, in which cancer does not arise in isolation but involves multiple cells simultaneously [22,23]. Therefore, it seems reasonable to analyse molecular changes in the tumour and the surrounding margin.

This is the first study to evaluate the levels of E2F2, MDM2 and p16 in the primary OSCC tumour and the matching margin samples and their association with the selected sociodemographic and clinicopathological characteristics.

## 2. Materials and Methods

### 2.1. Patient and Samples

This study included 73 Polish patients from whom the tumour (following the ICD-10 classification—C01: 42; C04.8: 24; C03.1: 7) and the matching margin tissue were obtained by surgical resection at the Department of Otorhinolaryngology and Oncological Laryngology, Faculty of Medical Sciences in Zabrze, Medical University of Silesia, Katowice. All the samples were analysed based on the criteria of the American Joint Committee on Cancer (AJCC, version 2007) [24,25] and WHO Classification of Head and Neck Tumours [26]. The margin tissues were confirmed as free from cancer by pathologists. The main inclusion criteria were as follows: the written informed consent to participate in the study, age over 18 years and the diagnosis of primary OSCC tumours. Patients with metabolic diseases (e.g., diabetes, hypertension), chronic inflammatory diseases and a history of preoperative radio- or chemotherapy were not included in the study. The project was approved by the Bioethics Committee of the Medical University of Silesia (approval No. KNW/022/KB1/49/16 and No. KNW/002/KB1/49/II/16/17). The research protocol is presented in Figure 1. The characteristics of the study group are shown in Table 1. The mean age was 61 years (range: 54–68 years).

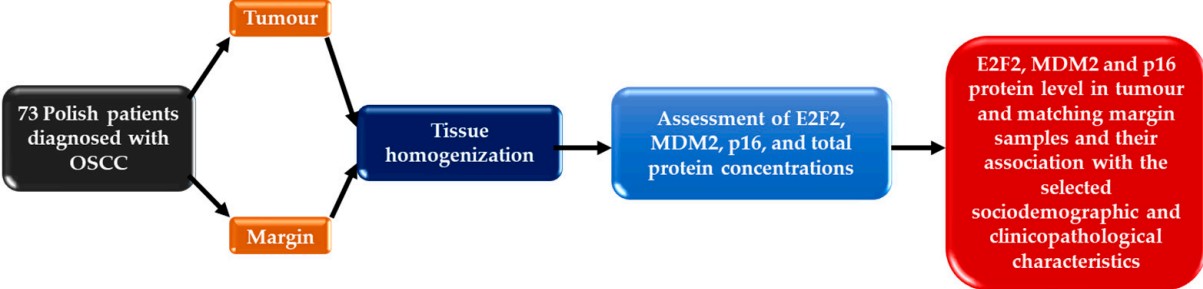

**Figure 1.** Diagram of the study protocol.

**Table 1.** Parameters characterizing OSCC patients.

| Parameters | Patients, n (%) |
|:---:|:---:|
| Gender | |
| Men | 56 (76.7) |
| Women | 17 (23.3) |
| Smoking | |
| Smokers | 49 (67.1) |
| Non-smokers | 24 (32.9) |
| Alcohol consumption | |
| Drinker | 46 (63) |
| Non-drinker | 27 (37) |
| HPV status | |
| HPV-positive | 18 (24.7) |
| HPV-negative | 47 (64.4) |
| No data | 8 (10.9) |
| Histological grading | |
| G1 (Well differentiated) | 15 (20.6) |
| G2 (Moderately differentiated) | 38 (52) |
| G3 (Poorly differentiated) | 20 (27.4) |
| T classification | |
| T1 | 12 (16.4) |
| T2 | 27 (37) |
| T3 | 24 (32.9) |
| T4 | 10 (13.7) |
| Nodal status | |
| N0 | 36 (49.3) |
| N1 | 9 (12.3) |
| N2 | 22 (30.2) |
| N3 | 6 (8.2) |

*2.2. E2F2, MDM2 and p16 Protein Concentration Evaluation*

All the analyses were performed at the Department of Medical and Molecular Biology, Faculty of Medical Sciences in Zabrze, Medical University of Silesia, Katowice. The samples were transported on ice to the department and then frozen at −80 °C until analysis. In the first step of the study, 10% homogenate in cold PBS (EURx, Gdańsk, Poland) was obtained from the tumour and the margin tissues using a homogenizer Bio-Gen PRO200 (PRO Scientific Inc., Oxford, CT, USA) at the rate of 10,000 RPM (5 times 1 min at 2 min intervals). The suspensions were sonicated with the ultrasonic processor UP100H (Hilscher, Teltow, Germany).

The enzyme-linked immunosorbent assay (ELISA) was used to determine E2F2 MDM2 and CDKN2A protein concentrations in sample homogenates (Assay ID: SEJ182Hu for E2F2; Assay ID: SEG790Hu for MDM2 and Assay ID: SEA794Hu for p16; Cloud-Clone Corp., Houston, TX, USA). All the tests were used according to the manufacturer's protocols. The measurements of absorbances were taken with a Synergy H1 microplate reader (BioTek, Winooski, VT, USA) and the results were calculated with Gen5 2.06 software (BioTek,

Winooski, VT, USA). The absorbance was read at the wavelength of 450 nm. The tests were characterized by the following sensitivities: 0.059 ng/mL for E2F2, 0.057 ng/mL for MDM2 and 0.262 ng/mL for p16. The intra-assay variation was below 10% and the inter-assays were below 12% for all the determined proteins. E2F2, MDM2 and p16 protein concentrations were normalized to the total amount of protein in the tissue homogenates and were expressed in ng/μg. The total protein was measured using an AccuOrange™ Protein Quantitation Kit (Biotium, Fremont, CA, USA) according to the standard instructions. The fluorescence was determined with excitation/emission at 480/598 nm (SYNERGY H1 microplate reader; BIOTEK, Winooski, VT, USA). The test detection was 0.1–15 μg/mL protein.

*2.3. HPV 16 Detection*

DNA was extracted from all the tissues with the use of a Gene Matrix Tissue DNA Purification Kit (EURx, Gdansk, Poland) according to the manufacturer's protocol. Following isolation, DNA concentration and quality were assessed using spectrophotometry in a Biochrom WPA Biowave DNA UV/Vis Spectrophotometer (Biochrom, Cambridge, UK). HPV was detected using an AmpliSens® HPV 16/18-FRT PCR kit (InterLabService, Moscow, Russia) according to the manufacturer's protocol and the QuantStudio 5 RealTime PCR System (Applied Biosystems, Foster City, CA, USA). The PCR was performed in a volume of 25 μL using 7 μL of PCR-mix-1-FEP/FRT HPV; 8 μL mixture of PCR-buffer-FRT and TaqF polymerase and 10 μL of DNA. The amplification program was as follows: 95 °C for 20 s, followed by 45 cycles of 95 °C for 20 s and 60 °C for 1 min.

*2.4. Statistical Analyses*

The Shapiro–Wilk test evaluated the distribution of the variables. The median with interquartile range (25–75%) was used to describe protein concentrations. The Mann–Whitney U test was used to compare the sociodemographic and clinical characteristics and protein concentrations where *p*-values < 0.05 were considered statistically significant. The statistical software STATISTICA version 13 (TIBCO Software Inc., Palo Alto, CA, USA) was used to perform all the analyses.

**3. Results**

We found no statistically significant differences in E2F2, MDM2 or p16 protein levels in tumour samples as compared to the margin samples (Table 2). The median protein concentrations of E2F2, MDM2 and p16 were higher in the tumour tissue than in the margin samples.

**Table 2.** E2F2, MDM2 and p16 protein concentrations in tumour samples compared to margin samples.

| | Protein Concentrations [ng/μg Protein] Me (Q1–Q3) | | |
| --- | --- | --- | --- |
| | **Tumour** | **Margin** | ***p*-Value** |
| E2F2 | 0.129 (0.056–0.247) | 0.082 (0.045–0.252) | 0.64 |
| MDM2 | 0.103 (0.033–0.215) | 0.049 (0.013–0.193 | 0.09 |
| p16 | 1.08 (0.714–1.51) | 0.859 (0.377–1.91) | 0.845 |

Me Median, Q1 lower quartile, Q3 upper quartile.

No association was detected between the protein concentration, age, gender, smoking, alcohol consumption, HPV status and TNM and G in the analysed tissues except for E2F2 and p16. Patients with N0 had a significantly lower E2F2 concentration than patients with N1 (0.125 vs. 0.373; *p*-value = 0.02) in the tumour samples. Higher E2F2 protein concentrations were also noted in patients with N3 compared to N0 in tumour samples. However, it was not statistically significant (0.237 vs. 0.125; *p*-value = 0.15). The respective results are shown in Figure 2.

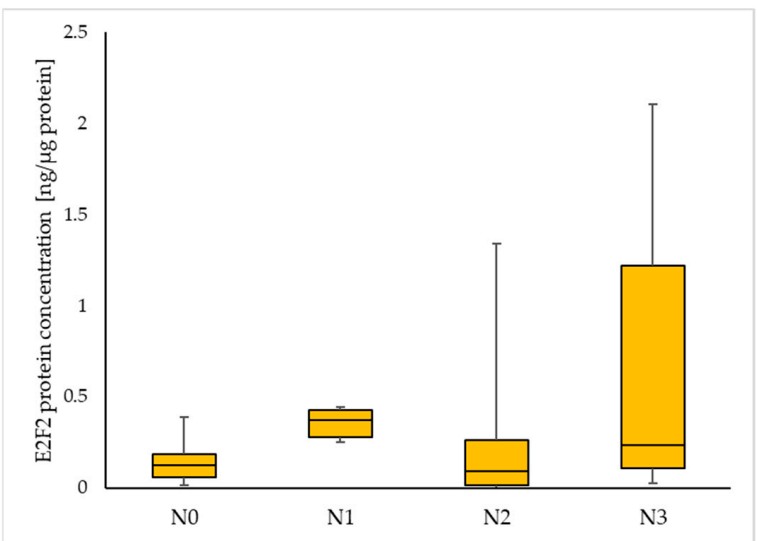

**Figure 2.** E2F2 protein concentration in the tumour samples according to N0, N1, N2 and N3.

Moreover, the median protein concentration of E2F2 was higher in HPV-negative patients in the tumour tissues (0.193 vs. 0.056; *p*-value = 0.03) compared to HPV-positive patients. Furthermore, alcohol drinkers had a lower level of p16 in the margin compared to non-drinkers (0.737 vs. 2.65; *p*-value = 0.01). Similar observations were found in concurrent drinkers and smokers compared to non-drinkers and non-smokers (0.745 vs. 5.965; *p*-value = 0.01). The results are given in Tables 3 and 4.

**Table 3.** E2F2, MDM2 and p16 protein concentration analyses in HPV-positive patients compared to HPV-negative patients.

| | | Protein Concentrations [ng/µg Protein] Me (Q1–Q3) | | |
|---|---|---|---|---|
| | | **HPV-Positive** | **HPV-Negative** | ***p*-Value** |
| E2F2 | Tumour | 0.056 (0.016–0.227) | 0.193 (0.107–0.330) | 0.03 |
| | Margin | 0.079 (0.025–0.18) | 0.0845 (0.05–0.394) | 0.37 |
| MDM2 | Tumour | 0.079 (0.033–0.254) | 0.104 (0.032–0.198) | 0.8 |
| | Margin | 0.052 (0.01–0.149) | 0.049 (0.014–0.198) | 0.82 |
| p16 | Tumour | 0.975 (0.379–1.576) | 1.094 (0.733–1.366) | 1.0 |
| | Margin | 1.0205 (0.426–1.513) | 0.752 (0.376–2.193) | 1.0 |

Me Median, Q1 lower quartile, Q3 upper quartile.

**Table 4.** E2F2, MDM2 and p16 protein concentration analyses in smokers vs. non-smokers; drinkers vs. non-drinkers; smokers and drinkers vs. non-smokers and non-drinkers.

| | | Protein Concentrations [ng/µg Protein] Me (Q1–Q3) | | |
|---|---|---|---|---|
| | | **Smokers** | **Non-Smokers** | ***p*-Value** |
| E2F2 | Tumour | 0.193 (0.065–0.365) | 0.12 (0.039–0.163) | 0.09 |
| | Margin | 0.075 (0.039–0.191) | 0.132 (0.059–0.356) | 0.31 |

**Table 4.** *Cont.*

| | | Protein Concentrations [ng/μg Protein] Me (Q1–Q3) | | |
|---|---|---|---|---|
| | | **Smokers** | **Non-Smokers** | ***p*-Value** |
| MDM2 | Tumour | 0.106 (0.035–0.234) | 0.079 (0.015–0.23) | 0.41 |
| | Margin | 0.033 (0.012–0.114) | 0.15 (0.021–0.285) | 0.23 |
| p16 | Tumour | 1.098 (0.716–1.687) | 1.094 (0.498–1.472) | 0.7 |
| | Margin | 0.755 (0.386–1.404) | 1.652 (0.329–5.265) | 0.23 |
| | | Drinkers | Non-drinkers | *p*-value |
| E2F2 | Tumour | 0.151 (0.062–0.249) | 0.125 (0.049–0.503) | 0.78 |
| | Margin | 0.072 (0.028–0.249) | 0.12 (0.068–0.267) | 0.35 |
| MDM2 | Tumour | 0.104 (0.0342–0.210) | 0.095 (0.019–0.237) | 0.62 |
| | Margin | 0.033 (0.011–0.159) | 0.15 (0.036–0.322) | 0.08 |
| p16 | Tumour | 0.967 (0.533–1.504) | 1.36 (1.072–2.548) | 0.09 |
| | Margin | 0.737 (0.338–1.219) | 2.65 (0.867–5.89) | 0.01 |
| | | Smokers and drinkers | Non-smokers and non-drinkers | *p*-value |
| E2F2 | Tumour | 0.177 (0.062–0.249) | 0.075 (0.031–0.122) | 0.05 |
| | Margin | 0.083 (0.035–0.297) | 0.298 (0.128–0.376) | 0.12 |
| MDM2 | Tumour | 0.104 (0.034–0.219) | 0.079 (0.010–0.30) | 0.6 |
| | Margin | 0.02 (0.011–0.080) | 0.152 (0.025–0.357) | 0.08 |
| p16 | Tumour | 0.975 (0.676–1.565) | 1.36 (1.153–1.51) | 0.29 |
| | Margin | 0.745 (0.387–1.159) | 5.965 (1.50–11.854) | 0.01 |

Me Median, Q1 lower quartile, Q3 upper quartile.

## 4. Discussion

E2F2, MDM2 and p16 are some of the key proteins associated with the control of the cell cycle [13,16,27]. However, the exact role of E2F2, MDM2 and p16 in prognosis and biological function has not been well established in OSCC. Our analysis showed no statistically significant differences in the expression levels of E2F2, MDM2 or p16 proteins in the tumour samples compared to the margin samples. Of note, patients with N0 had significantly lower E2F2 levels than patients with N1 nodal status (0.125 vs. 0.373; *p*-value = 0.02). Importantly, this observation was connected with the tumour tissues. Perhaps such results are related to the fact that E2F2 can promote cell division [28]. In addition, E2F family transcription factors are relevant components of the Rb suppressor protein pathway. The occurrence of alterations in the form of deletions, mutations, promoter methylation or amplification of genes encoding proteins from the E2F family of transcription factors is often reported in human cancers [29]. Some studies showed increased E2F2 expression in many cancer types at both mRNA and protein levels (hepatocellular carcinoma, cervical cancer, ovarian cancer, non-small cell lung cancer, breast cancer, bladder cancer and kidney cancer), which was often associated with a higher tumour grade, lymph node metastasis, lymph node invasion and a worse patient prognosis [29–32]. In addition, reduced E2F2 concentrations were observed in renal cell carcinoma, which was associated with decreased cell proliferation and invasion [33]. On the other hand, Shang et al. [34] reported that low E2F2 protein levels in colorectal cancer were correlated with the lymph node status, metastasis and pathological stage of the tumour [34]. Thus, it can be hypothesized that dysregulated E2F may mediate tumorigenic processes at multiple levels, including mediating DNA damage response, apoptosis, angiogenesis, genomic stabilisation and metabolism. The function of E2F family transcription factors is complicated as they can act as tumour suppressors or oncogenes, depending on the context or interaction environment. Liu et al. [35] demonstrated that E2F2 could have a role in pancreatic cancer cells by promoting the cell cycle transition from the G1 to the S phase, thereby increasing tumour cell proliferation [35]. Similar reports were confirmed in ovarian cancer, where E2F2 was identified as "a transcription factor that promotes proliferative processes" [31]. In the context of research on the potential use of

the E2F2 protein as a biomarker of OSCC progression, it seems important to analyse the factors that could modify its concentration. In this study, we demonstrated that the median protein concentration of E2F2 was higher in HPV-negative patients in the tumour tissues (0.193 vs. 0.056; *p*-value = 0.03). These results are not related to the evidence that the HPV oncogene E7 can bind to retinoblastoma (Rb) family proteins and precisely control E2F2. Ubiquitination of the Rb protein leads to the release of E2F factors. These factors transcribe cyclin E, cyclin A and p16, an inhibitor of CDK4/6, forcing cells to prematurely enter the S phase [36]. Perhaps the findings of our analysis indicates the possible presence of other factors besides the E7 oncogene that could affect the level of E2F2.

The next protein we analysed was MDM2. The median protein concentrations of MDM2 were higher in the tumour tissue than in the margin samples. However, they were not statistically significant. No association was detected between MDM2 protein concentration, TNM and G in the analysed tissues. In non-transformed cells, MDM2 and p53 proteins are weakly expressed, while MDM2 upregulation in cancer cells can be, in many cases, induced not only by p53-independent factors but also caused by the induction of p53-independent pro-survival mechanisms, including inhibition of tumour suppressor activity of the Rb protein or E2F transcription factor 1 [37]. The elevated MDM2 level was found in many cancers, such as lung cancer, breast cancer, liver cancer, oesophagogastric cancer, colorectal cancer, sarcomas, osteosarcomas, gliomas, melanomas and hematopoietic malignancies [16,38]. In human breast cancer, the MDM2 protein concentration was detected as a prognostic biomarker [37]. Importantly, MDM2 was more frequently overproduced in metastatic and recurrent tumours compared to primary tumours [39]. In most cancers, higher MDM2 concentrations were associated with a poorer clinical prognosis and a more advanced stage of the disease [40]. MDM2 overexpression protein has also been reported in HNSCC and laryngeal cancer [41–43]. Modification in MDM2 concentration is also related to a poorer prognosis in squamous cell carcinoma of the tonsillar region [44].

In this study, the median protein concentrations of p16 were higher in the tumour tissue than in the margin samples. However, they were not statistically significant. It was demonstrated that compared to non-drinkers, alcohol drinkers showed p16 protein expression lower in the margin (0.737 vs. 2.65; *p*-value = 0.01). Similar observations were found in concurrent drinkers and smokers compared to non-drinkers and non-smokers (0.745 vs. 5.965; *p*-value = 0.01). Changes in the gene expression profile are known to be crucial in the neoplastic transformation of healthy cells, and the best studied non-genetic risk factor is tobacco smoking. The components of tobacco smoke contain mutagenic substances, which may account for the altered expression of p16 protein in smokers. This is consistent with the results by Rungraungrayabkul et al. [45], who explained a change in p16 level due to stimulants, including smoking and/or alcohol consumption that could lead to changes in the form of deletion or methylation of the gene promoter and result in the loss of p16 expression [45]. In addition, there are prerequisites in smokers with p16 concentration pointing to tumours developing earlier than in non-smokers, and with a tendency for a higher tumour grade [46]. Contrary to our results, using immunohistochemistry (IHC), Belobrov et al. [47] examined p16 in formalin-fixed specimens (FFPE) to observe the significant overexpression of this protein in non-smokers and in non-alcohol drinkers [47]. These discrepancies could be due to such factors as differences in sampling techniques, preparation and detection methods. In our study, the median concentration of p16 in the study group was higher in the tumour tissue than in the margin samples, which seemed to confirm the results of studies carried out with the use of the bioinformatics instruments and IHC [48–52]. Interestingly, it has been speculated that the p16 concentration could be higher in OSCC patients with HPV infection [53]. The correlation is attributed to the effect of the viral protein E7 that is able to inhibit tumour suppressors, particularly Rb1 protein, which could result in an increased p16 level [47,54]. Consequently, multiple studies have attempted to evaluate the utility of p16 as a surrogate marker for HPV infection in OSCC. A large cross-sectional study found no HPV DNA or E6 mRNA in 1260 OSCC samples [51]. Similar to this study,

other authors reported no association between the level of p16 and HPV infection, either during virus detection performed by ISH or PCR [45,47,55–59]. On the other hand, an Asian study suggested that the correlation between p16 and the HPV status in OSCC varied according to the tumour location [56]. This could be due to the fact that the oral cavity is anatomically a diverse region, and recent studies have identified OSCC as a heterogeneous group of cancers in terms of their anatomy, histology and microenvironment [56,60,61]. On the other hand, several research teams confirmed the usefulness of IHC p16 as a surrogate biomarker of HPV, while Berdugo et al. [62] emphasized that overexpression of the p16 protein could also be independent of the presence of the virus. However, it could also result from changes in Rb1 protein, p53 or CDK6 [62,63].

In summary, the small sample size is the main limitation of this study. Our results should be validated on larger and diverse cohorts. In addition, such an analysis should also include the cell lines to better understand the role of these proteins in oral carcinogenesis.

## 5. Conclusions

E2F2 could potentially promote tumour progression and metastasis. Moreover, our results showed a differential level of the analysed proteins in response to alcohol consumption and the HPV status.

**Author Contributions:** K.G. and A.Ś., writing of the article; K.G., research concept and design; K.M.-O. and N.Z., collection of samples; A.Ś., K.G. and J.G., protein level analyses; K.G., DNA extraction; K.B., D.H. and A.Ś., HPV detection; K.G. and D.H., data analysis and interpretation; M.M. and J.K.S., critical revision of the article. All authors have read and agreed to the published version of the manuscript.

**Funding:** This research was funded from the grant by the Medical University of Silesia (PCN-2-028/N/1/O).

**Institutional Review Board Statement:** This study was conducted according to the guidelines of the Declaration of Helsinki, and approved by the Bioethics Committee of the Medical University of Silesia (approval No. KNW/022/KB1/49/16 and No. KNW/002/KB1/49/II/16/17).

**Informed Consent Statement:** Informed consent was obtained from all subjects involved in this study.

**Data Availability Statement:** The data used to support the findings of this study are available from the corresponding author upon request.

**Conflicts of Interest:** The authors declare no conflict of interest. The funders had no role in the design of this study, in the collection, analyses, or interpretation of data, in the writing of the manuscript or in the decision to publish the results.

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
