# Peer review of "The Potential Association between E2F2, MDM2 and p16 Protein Concentration and Selected Sociodemographic and Clinicopathological Characteristics of Patients with Oral Squamous Cell Carcinoma"

_cimb, doi:10.3390/cimb45040213_

Round 1
Reviewer 1 Report
Comments and Suggestions for Authors
Thanks for the opportunity to review this manuscript about the Potential Association Between E2F2, MDM2 and p16 Protein Concentration and Selected Sociodemographic and Clinicopathological Characteristics of Patients with Oral Squamous Cell Carcinoma.
The manuscript is well written but there are many concerns in methodology and results.
· The introduction is well written.
Materials and Methods
· The methodology is obscure, and much information is not clear or missing.
· What is type of OSCC used? recurrence or primary? Location in oral cavity? did you include cases with metastasis?
· Why the researchers used margins as control? What is the status of these margins?
· What is the source of sample: fresh, Frozen, and FFPE?
· How the researchers extract proteins?
· Further tests are highly recommended for protein expression like IHC.
· The company name and country for reagent, software, or equipment should be mentioned.
Results:
· The tables are not well organized.
· The association between tumor expression and N category is not clear? How correlated? Did the researchers check the expression in positive LN?
Discussion:
· Many irrelevant information that is not supporting the findings.
· Please mention the limitation of the study
· What is the clinically relevant of this study.
Reviewer 2 Report
Comments and Suggestions for Authors
Review on ÅšwiÄ™tek et al’s “The Potential Association Between E2F2, MDM2 and p16 Protein Concentration and Selected Sociodemographic and Clinicopathological Characteristics of Patients with Oral Squamous Cell Carcinoma”
The study is understandable, well displayed. Basically, the statistical analysis is fine, though with other, a bit more sophisticated statistical methods more could have been said about the relationships between the expressions of the proteins and the clinical characteristics.
Which post hoc test was exactly used in case of Kruskal-Wallis tests? Please, include to statistical methods!
Round 2
Reviewer 1 Report
Comments and Suggestions for Authors
Thank you for answering the comments.
Author Response
Dear Reviewer 1,
We do not understand the Reviewer's decision.
In the first round, we answered all the Reviewer's questions. Reviewer 1 did not indicate why the research design appropriate must be improved.
In the second round, we only received the comment 'Thank you for answering the comments'. It is therefore impossible to do the major revision recommended by Reviewer 1.